# The Southern Ocean diatom *Pseudo-nitzschia subcurvata* flourished better under simulated glacial than interglacial ocean conditions: Combined effects of $CO_2$ and iron

**Anna Pagnone**, **Florian Koch***, **Franziska Pausch**, **Scarlett Trimborn**

EcoTrace, Alfred Wegener Institute, Helmholtz Centre for Polar and Marine Research, Bremerhaven, Germany

* florian.koch@awi.de

**Data Availability Statement:** All relevant data are within the manuscript and its Supporting Information files.

## Abstract

The 'Iron Hypothesis' suggests a fertilization of the Southern Ocean by increased dust deposition in glacial times. This promoted high primary productivity and contributed to lower atmospheric $pCO_2$. In this study, the diatom *Pseudo-nitzschia subcurvata*, known to form prominent blooms in the Southern Ocean, was grown under simulated glacial and interglacial climatic conditions to understand how iron (Fe) availability (no Fe or Fe addition) in conjunction with different $pCO_2$ levels (190 and 290 µatm) influences growth, particulate organic carbon (POC) production and photophysiology. Under both glacial and interglacial conditions, the diatom grew with similar rates. In comparison, glacial conditions (190 µatm $pCO_2$ and Fe input) favored POC production by *P. subcurvata* while under interglacial conditions (290 µatm $pCO_2$ and Fe deficiency) POC production was reduced, indicating a negative effect caused by higher $pCO_2$ and low Fe availability. Under interglacial conditions, the diatom had, however, thicker silica shells. Overall, our results show that the combination of higher Fe availability with low $pCO_2$, present during the glacial ocean, was beneficial for the diatom *P. subcurvata*, thus contributing more to primary production during glacial compared to interglacial times. Under the interglacial ocean conditions, on the other hand, the diatom could have contributed to higher carbon export due to its higher degree of silicification.

## Introduction

The Southern Ocean (SO) is the world's largest high-nutrient low-chlorophyll region (HNLC) and an area where physical forcing, atmospheric $pCO_2$, biological production and marine biogeochemical cycles are tightly linked. In this region, primary production is restricted by the bioavailability of the trace metal (TM) iron (Fe) [1–3]. Fe is an essential trace element, which is needed by phytoplankton to transfer electrons in key cellular and metabolic processes including photosynthesis, respiration, chlorophyll production, carbon (C) and nitrogen (N) fixation [4]. The availability of Fe strongly influences phytoplankton species composition and growth [5–9], and impacts the biological carbon pump and thus the global carbon cycle.

**Funding:** AP was supported by German Federal Ministry of Education and Research (BMBF) as Research for Sustainability initiative (FONA); www. fona.de through Palmod project (FKZ: 01LP1505C). ST, FK and FP were funded by the Helmholtz Association (HGF Young Investigators Group EcoTrace, VH-NG-901). The funders had no role in study design, data collection and analysis, decision to publish, or preparation of the manuscript.

**Competing interests:** The authors have declared that no competing interests exist.

Consequently, changes in Fe availability have caused feedback effects on climate over geological timescales. Furthermore, the SO is a region of high $CO_2$ exchange between ocean and atmosphere [10, 11]. The SO has been reported to be a major sink of atmospheric $CO_2$ during glacial periods, while it was on the other hand a source of $CO_2$ during glacial-interglacial transitions. At present day, the SO is the major sink of anthropogenic $CO_2$ [12, 13].

During the Last Glacial Maximum (LGM), the SO experienced changes in oceanic circulation and carbon storage. For instance, increased sea ice extent strengthened surface water stratification, thus limiting ocean ventilation and trapping more carbon in the deep ocean [14–16]. Additionally, the northward displacement of the westerly winds prevented the upwelling of $CO_2$-rich deep water [17]. Besides physical mechanisms, the strength of the biological pump might explain 25–50% of the roughly 100 µatm $pCO_2$ discrepancy between glacial (180 µatm $pCO_2$) and interglacial (280 µatm $pCO_2$) times as argued in several studies [18–21]. This supports John Martin's 'Iron Hypothesis', which suggests that an increase in dust deposition during glacial times would fertilize the ocean, stimulate marine productivity, and enhance C export [1, 2, 22–24]. Indeed, analysis of sediment cores revealed a positive correlation between aeolian Fe supply and primary production during ice ages [20, 25]. Proxy data as well as model simulations showed a doubling of the global dust deposition during the last glacial climate condition, when 826 Tg/yr dust were deposited in the global ocean, compared to the 440 Tg/yr dust in pre-industrial times [26, 27]. The difference was mainly due to a dryer atmosphere and reduced vegetation cover [22]. Used a biogeochemical model to estimate the impact of Fe deposition on the global ocean. Under current conditions, 33% of the world's oceans water masses have Fe concentrations, which limit the growth of phytoplankton. The model simulations revealed that the percentage of Fe-poor water masses decline to 25% and 13% with pre-industrial and LGM dust input, respectively. Along with the North Pacific Ocean, the SO showed the most significant difference in soluble Fe deposition during glacial and interglacial times, accordingly having the largest impact on marine biogeochemistry [22]. The dust deposition in the SO during glacial times was roughly ten times higher (0.04–0.17 Tg/yr) than in pre-industrial times (0.005–0.018 Tg/yr) [27]. However, the SO is geographically isolated from arid, dust-producing regions and is thus overall characterized by low aeolian Fe deposition [28]. Other sources of Fe include upwelling of deep nutrient-rich water, entrainment of sedimentary Fe from continental shelfs and resuspension, island-wake effects, seasonal sea ice extent and melt, as well as iceberg drift and melt [3 and references therein].

The phytoplankton community in the current SO is dominated by different diatom species and the prymnesiophyte *Phaeocystis antarctica* [29]. Diatoms account for 40% of the ocean's total primary production [30–33] and dominate the export of particulate organic matter to the seafloor [34, 35]. In other words, diatoms are crucial for the ocean's ability to sequester C to the ocean's interior. Diatoms also have an extensive impact on the oceanic silica inventory, as they produce frustules containing silica. Some frustules are resistant to remineralization and dissolution, are well preserved in the sediment, and thus provide precious information about past oceanic biogeochemistry. *Pseudo-nitzschia* species have been frequently observed in today's phytoplankton assemblages in Antarctic waters [36]. Mesoscale Fe fertilization experiments in the SO triggered massive phytoplankton blooms dominated by large diatoms like the pennate *Pseudo-nitzschia* sp. [7, 37]. Large diatoms in the SO appear to have a higher Fe requirement compared to smaller phytoplankton because of physical constrains in the Fe uptake process [38]. To compensate for this, they have evolved various strategies to acquire bioavailable Fe. They generally reduce their biogeochemical Fe requirement through metal or protein substitution [39] and reduce Fe-rich components of the photosystem apparatus [9]. [8] suggested that *Pseudo-nitzschia* is able to accumulate intracellular Fe when ambient concentrations of this TM are high, while maintaining a low Fe demand. This luxury uptake and

subsequent storage of Fe supports growth in subsequent low Fe environments and enables *Pseudo-nitzschia* to dominate phytoplankton assemblages across a wide range of oceanic Fe concentrations.

In SO diatoms, Fe limitation often results in slower growth and reduced C fixation. The photochemical quantum efficiency, which indicates how efficiently excitation energy is transferred to the reaction centers, is usually lowered [8, 40–42]. In an Fe-poor environment, cells usually increase the functional absorption cross sectional area of their reaction centers, thereby enhancing the target area, which absorbs incoming photons [38, 41]. The absorbed photons can either drive photosynthesis, N reduction, C fixation, photorespiration or can be converted to heat (non-photochemical quenching). Fe deficiency induces changes in the photosystem II (PSII) reaction centers such as the reduction of the pigment content [43], causes less efficient electron transport [40] and increases non-photochemical quenching to dissipate the excess light energy [44].

Besides Fe limitation, phytoplankton cells have experienced variations in CO$_2$ concentration in the past. Previous studies on the effect of high CO$_2$ concentrations on phytoplankton reported changes in their elemental composition (e.g. [45, 46]), in cell size (e.g. [47]) and in the degree of silicification in diatoms (e.g. [48]). Furthermore, it was shown that low pCO$_2$ levels can influence the composition of Antarctic phytoplankton communities. For example, experiments with natural phytoplankton assemblages from different regions across the SO [24, 49, 50] concluded that *Pseudo-nitzschia* flourishes at low pCO$_2$ levels, while it does not do well in response to ocean acidification. Indeed, between ambient and future elevated pCO$_2$ levels, the growth of *P. subcurvata* in a laboratory experiment was not stimulated under enhanced Fe supply [51]. Under similar Fe conditions, a phytoplankton community from the Ross Sea, Antarctica, responded to CO$_2$ increase from 100 to 800 ppm with a dramatic reduction in cell abundance of *P. subcurvata*, being replaced by *Chaetoceros* species [49]. Similarly, a community from the Weddell Sea, Antarctica, shifted from *Pseudo-nitzschia* to *Fragilariopsis* after Fe addition between 390 to 800 µatm pCO$_2$ [50], while no difference in species composition was found between the glacial (190 µatm) and the present-day (390 µatm) pCO$_2$ levels. This implies that reduced CO$_2$ concentrations during glacial periods potentially favored pennate diatoms such as *Pseudo-nitzschia* while diatom species such as *Chaetoceros* and *Fragilariopsis* became most abundant under present-day and future pCO$_2$ levels [49]. A few studies investigated the SO phytoplankton assemblages and growth under low Fe supply in response to increasing pCO$_2$ [24, 50, 52]. [50] observed also a CO$_2$-dependent taxonomic shift in Fe-deplete conditions with increasing pCO$_2$ with *Pseudo-nitzschia* being replaced by the pennate diatom *Synedropsis* between 390 and 800 µatm pCO$_2$ levels. Similarly, when pCO$_2$ increased from 390 to 900 µatm another SO plankton community changed from being dominated by *P. prolongatoides* to one, which was dominated by *P. antarctica* [24]. Hence, irrespective of Fe availability the genus *Pseudo-nitzschia* was found to be susceptible to ocean acidification pCO$_2$ levels.

Studies that asses the effects of low pCO$_2$ on phytoplankton often compare their results with high pCO$_2$ levels to understand ocean acidification. However, little is known about the smaller variation from 180 (glacial) to 280 µatm (interglacial/pre-industrial) pCO$_2$ under different Fe availability. Indeed, the potential interactive effect of low-pCO$_2$ (180 and 280 µatm) together with different Fe availability (deplete and replete) on net primary production and export production is currently often not considered, when developing models or designing laboratory experiments simulating glacial and interglacial ocean conditions. Studies looking at N-isotopes and Th-corrected sediment accumulation rates describe large fluxes of biogenic detritus out of surface waters in the glacial ocean due to a larger amount of lithogenic Fe transported by winds [26]. The latter study indicates that increased export production and thus

enhanced C storage potentially contributed to the observed lower atmospheric $CO_2$ concentrations during glacial times [53].

The above-mentioned studies offer first insights on how some phytoplankton species cope with glacial and interglacial climatic conditions. However, studies on the ecophysiology of Antarctic diatoms subject to glacial vs. interglacial ocean conditions under reduced Fe conditions, are yet lacking. In this study, the SO bloom-forming diatom *P. subcurvata* was grown under Fe and $CO_2$ conditions representative of glacial (lower $CO_2$ and higher Fe) and interglacial (higher $CO_2$ and lower Fe) times to untangle the influence of these two environmental factors on growth, elemental stoichiometry, photosynthetic carbon production and photophysiology. This allowed to assess its role in the paleo carbon cycle.

## Material and methods

### Experimental setup

Prior to the execution of the experiment, the oceanic diatom *P. subcurvata* (isolated by Philipp Assmy at 49˚S, 2˚E, R/V Polarstern cruise ANT-XXI/4, April 2004) was grown for more than one year in Antarctic seawater with a low total dissolved Fe (dFe) concentration of 0.5 nmol L$^{-1}$ Preacclimation and the main experiment were carried out in Fe-poor (0.4 nmol L$^{-1}$) Antarctic seawater collected at 60˚32S, 26˚29W (salinity of 33.8 ± 0.2), filtered through a sterilized, acid-cleaned 0.2 μm filter (Sartobran, Sartorius). This water was spiked with chelexed (Chelex® 100, Sigma Aldrich, Merck) macronutrients (100 μmol L$^{-1}$ Si, 100 μmol L$^{-1}$ NO$_3^-$ and 6.25 μmol L$^{-1}$ PO$_4^{3-}$) and vitamins (30 nmol L$^{-1}$ B$_1$, 23 nmol L$^{-1}$ B$_7$ and 0.228 nmol L$^{-1}$ B$_{12}$) according to the F/2$_R$ medium [54]. In addition, a TM mix containing Zn (0.16 nmol L$^{-1}$), Cu (0.08 nmol L$^{-1}$), Co (0.09 nmol L$^{-1}$ Co), Mn (1.9 nmol L$^{-1}$), Mo (0.05 nmol L$^{-1}$) in the ratio of the original F/2 recipe adjusted to 4 nmol L$^{-1}$ Fe was added. As suggested by [55], in order to minimize the alteration of the natural seawater TM chemistry and ligands, no ethylenediaminetetraacetic acid (EDTA) was added. The Fe-deplete treatments (henceforth referred to as *Control*) contained 0.4 nmol L$^{-1}$ dFe while for the Fe-enriched treatments (henceforth referred to as *+Fe*), 4 nmol L$^{-1}$ FeCl$_3$ were added.

To avoid Fe contamination, TM clean techniques were used according to the GEOTRACES cookbook [56]. The sampling and handling of the incubations was conducted under a laminar flow hood (Class 100, Opta, Bensheim, Germany). All equipment was soaked for one week in 1% Citranox, followed by two weeks in 1 N HCl for polycarbonate and 5 N HCl for polyethylene materials. In between and after the cleaning process, the equipment was rinsed seven times with Milli-Q (MQ, Millipore). Finally, everything was air dried under a clean bench (U. S. class 100, Opta, Bensheim, Germany) and packed in three polyethylene bags.

All *Control* and *+Fe* incubations were bubbled with humidified air containing pCO$_2$ levels of 190 and 290 μatm, henceforth referred to as *190* and *290*, respectively. Using a gas flow controller (CGM 2000, MCZ Umwelttechnik, Bad Nauheim, Germany), both $CO_2$ gas mixtures were generated by combining $CO_2$ free air (< 1 ppmv $CO_2$, Dominick Hunter, Kaarst, Germany) with pure $CO_2$ (Air Liquide Deutschland Ltd., Düsseldorf, Germany) in the respective ratios. They were regularly monitored with a Li-Cor (LI6252 Biosciences, Lincoln, NE) calibrated with $CO_2$ free air and purchased gas mixtures of 150 ± 10 and 1000 ± 20 ppmv $CO_2$ (Air Liquide Deutschland Ltd., Düsseldorf, Deutschland). Low pCO$_2$ and Fe input characterized the glacial ocean, which was here simulated in the *+Fe 190* treatment. Vice versa, the interglacial ocean was characterized by higher pCO$_2$ and no Fe input and mimicked by the *Control 290* treatment. In addition to the incubation bottles, Fe and carbonate chemistry were determined in the culture medium which was incubated in the same way as the respective

incubation bottles ($pCO_2$ and Fe availability), to check if the different $pCO_2$ and Fe manipulations were successful.

All incubations were placed in front of LED (light-emitting diode) lamps at 100 μmol photons m$^{-2}$ s$^{-1}$ under a light:dark cycle of 16:8 h. The light intensity was adjusted with a LI-1400 datalogger (Li-Cor Biosciences, Lincoln, NE, USA) with a 4π-sensor (Walz, Effeltrich, Germany). For this experiment, the long-term low Fe acclimated *P. subcurvata* stock culture was inoculated to the different $CO_2$-Fe conditions and was acclimated to each experimental condition at 2˚C for at least two weeks. The main experiment was carried out in triplicate 4 L acid-cleaned polycarbonate bottles for each experimental treatment. The main experiment started with initial cell densities of ~1000 cells mL$^{-1}$, lasted between 8 and 9 days and reached final cell densities between 67 000 and 107 000 cells mL$^{-1}$.

## Trace metal chemistry

At the end of the experiment, total dissolved Fe (dFe) samples were taken from the culture medium by filtering 100 mL from each bottle through 0.2 μm HCl-cleaned polycarbonate filters (47 mm, Nuclepore, Whatman, GE Healthcare, Chicago, IL, USA) using a trace metal clean filtration system under a clean laminar flow hood (Class 100, Opta, Bensheim, Germany). The filtrate was then filled into a 125 ml HCl-cleaned PE bottle and stored triple-bagged at 2˚C until analysis. Between each filtration, the filtration manifold was cleaned in an acid bath consisting of 1 M HCl and rinsed seven times with Milli-Q. Prior to the dFe analysis, 0.2 μm pre-filtered seawater samples were acidified to pH 1.75 with double distilled $HNO_3$, minimizing the formation of Fe and Mn hydroxides. Next, samples were UV (ultraviolet) oxidized for 1.5 h using a 450 W photochemical UV power supply (photochemical lamp 7825; Power Supply 7830, ACE GLASS Inc., Vineland N.J., USA). Total dFe concentration of the seawater samples and the processed blanks were measured with a seaFAST system (Elemental Scientific, Omaha, NE, USA) [57] coupled to a sector field inductively coupled plasma mass spectrometer (ICP-MS; Element 2, Thermo Fisher Scientific; resolution of R = 4000; oxide forming rates below 0.3%). To minimize matrix effects, the seawater dFe concentrations were analyzed by standard addition. The accuracy of the dFe data was assessed by measuring NASS-6 (National Research Council of Canada) reference standards, with a recovery rate for Fe of 110%.

## Carbonate chemistry

From the culture medium as well as from the incubation bottles at the end of the experiment, dissolved inorganic carbon (DIC) was filtered through 0.2 μm filters (Nalgene, Thermo Scientific) and was stored at 4˚C in 5 mL borosilicate glass bottles without headspace. The colorimetric analysis was performed with a QuAAtro autoanalyzer (Seal Analytical, [58]). Again, from the culture medium as well as from the incubation bottles at the end of the experiment, samples for the total alkalinity (TA) were filtered through 0.6 μm GF/F filters (Whatman) and stored at 4˚C in 150 mL borosilicate glass bottles. TA was measured via potentiometric titration [59] and the concentrations were calculated using a linear Gran Plot [60]. The $pCO_2$ was calculated using the CO2Sys program [61] with the equilibrium constants of [62] as refitted by [63] using TA and DIC measurements, concentrations of phosphate and silicate, temperature and salinity.

## Growth

Cell count samples of *P. subcurvata* were fixed with 10% acid lugol solution and stored at 2˚C in the dark until counting. Cell numbers of *P. subcurvata* were enumerated according to the

method by [64] using 3 ml sedimentation chambers (Hydrobios, Kiel, Germany) on an inverted microscope (Zeiss Axiovert 200) counting at least 400 cells.

The growth rates μ ($d^{-1}$) were determined with

$$\mu = \frac{\ln\left(\frac{N_t}{N_0}\right)}{\Delta t}$$

where $N_0$ and $N_t$ denote the initial and the final cell concentrations of the experiments, respectively and $\Delta t$ is the incubation time in days. Final harvest took place when the cells were in exponential growth and reached densities between 67 000 and 107 000 cells $mL^{-1}$.

The cell volume was computed using the volume formula of a prism on parallelogram base provided by [65]. The apical and transapical axes were measured via microscopy, while the pervalvar axis was estimated to be half of the transapical axis with an average value of 1.2 μm.

## Elemental composition

At the end of the experiment, particulate organic carbon (POC) and particulate organic nitrogen (PON) were measured after filtering onto pre-combusted (15 h, 500°C) GF/F filters (pore size ~ 0.6 μm, Whatman). The amount of seawater filtered ranged between 200–300 mL and was dependent on the biomass in the treatments. Filters were stored at -20°C and dried for > 12 h at 60°C. Analysis was performed using a Euro Elemental Analyzer 3000 CHNS-O (HEKAtech GmbH, Wegberg, Germany). At the end of the experiment, samples to determine biogenic silica (BSi) were filtered through a cellulose acetate filter (Sartorius, 0.6 μm) and stored at -20°C. The dried filters were submerged in 0.2 M NaOH at 95°C for 45 minutes, cooled in an ice bath for 15 minutes, neutralized with 1 M HCl according to [66] and analyzed colorimetrically for silicate using standard spectrophotometric techniques [67]. Contents of POC, PON and BSi were corrected for blank measurements and normalized to filtered volume and cell densities to obtain cellular quotas. Production rates of POC, PON and BSi were calculated by multiplying the cellular quotas with the respective growth rate.

## Pigments

The amount of seawater filtered to collect pigment ranged between 200–300 mL on the GF/F filter and was dependent on the biomass in the treatments. Each pigment sample was flash frozen in liquid nitrogen and stored at -80°C until analysis. First, the pigments were homogenized and extracted for 24 h in 90% acetone at 4°C in the dark. Second, they were centrifuged for five minutes (4°C, 13000 rpm) and filtered through a 0.45 μm pore size nylon syringe filter (Nalgene, Nalge Nunc International, Rochester, NY, USA). The pigments were analyzed by reversed phase High Performance Liquid Chromatography (HPLC) on a LaChromElite system equipped with a chilled autosampler L-2200 and a DAD detector L- 2450 (VWR-Hitachi International GmbH, Darmstadt, Germany). A SpherisorbODS-2 column (25 cm × 4.6 mm, 5 μm particle size; Waters, Milford, MA, USA) with a LiChropher100-RP-18 guard cartridge was used for the separation of pigments, applying a gradient according to [68]. Peaks of light harvesting (LH) pigments chlorophyll *a* (Chl *a*) and $c_2$ (Chl $c_2$), fucoxanthin (Fuco), as well as of the light protective (LP) pigments diatoxanthin (Dt) and diadinoxanthin (Dd) were detected, identified and quantified by co-chromatography with the corresponding pigment standards (DHI Lab Products, Horsholm, Denmark) using the software EZChrom Elite ver. 3.1.3. (Agilent Technologies, Santa Clara, CA, USA). Pigment contents were normalized to filtered volume and cell densities to obtain cellular quotas.

## Photophysiological parameters

The efficiency of photochemistry in the PSII of *P. subcurvata* was assessed regularly during and at the end of the experiment by means of a Fast Repetition Rate fluorometer (FRRf, FastOcean PTX) and a FastAct Laboratory system (both from Chelsea Technologies Group ltd., West Molesey, United Kingdom). Values were obtained using the FastPro8 software (Version 1.0.50), [69]. Measurements were performed at least 2 hours after begin of the light period at 2°C after 10 minutes of dark-adaptation to ensure that all PSII reaction centers were fully oxidized and non-photochemical quenching (NPQ) was relaxed [70]. For each treatment, a 0.2 μm filtered blank was collected, measured and subtracted.

The fluorometer's LED (wavelength 450 nm) was automatically adjusted to a light intensity of $1.2 \cdot 10^{22}$ photons $m^{-2}$ $s^{-1}$. A single turnover flashlet was applied to cumulatively saturate PSII, thus to close all PSII reaction centers, and consisted of 100 flashlets on a 2 μs pitch, followed by a relaxation phase made of 40 flashlets on a 50 μs pitch to reopen the PSII reaction centers. The saturation phase of the single turnover acquisition, comprised 24 sequences and was fitted according to [71]. The minimum ($F_0$) and maximum ($F_m$) Chl *a* fluorescence were determined and the apparent maximum PSII quantum yield ($F_v/F_m$) was calculated according to the equation:

$$F_v/F_m = (F_m - F_0)/F_m$$

Further outputs of the FastPro8 software from the single turnover measurements of dark-adapted cells were the connectivity between PSII (P, dimensionless), thus the energy transfer between PSII units, the time constant for electron transport at the acceptor side of PSII ($\tau$, μs), the functional absorption cross section of PSII photochemistry ($\sigma_{PSII}$, $nm^{-2}$) and the cellular concentration of functional PSII reaction centers (RCII, zmol $cell^{-1}$).

During the photosynthesis-irradiance-curve (PE-curve), cells were exposed to eight light levels ranging from 0 to 1868 μmol photons $m^{-2}$ $s^{-1}$ for five minutes each. At each light level, six measurements of the light-adapted minimum ($F'$) and maximum ($F_m'$) Chl *a* fluorescence were taken and the effective PSII quantum yield ($F_q'/F_m' = (F_m'-F')/F_m'$) was calculated [72].

Cellular electron transport rates (cETR) were calculated following [73, 74] and normalized by RCII [75] using:

$$cETR = RCII \cdot \sigma_{PSII} \cdot E \cdot \frac{F_q'/F_m'}{F_v/F_m}$$

where E (photons $m^{-2}$ $s^{-1}$) is the applied instantaneous irradiance, which was measured separately for each light level in seawater.

The cETR versus E curve was fitted according to [76] allowing to derive the maximum cETR ($cETR_{max}$), the minimum saturating irradiance ($I_K$) determined by the interception of the light-limited region with the maximum photosynthetic rate, and the maximum light utilization efficiency ($\alpha$).

NPQ of Chl *a* fluorescence was calculated using the Stern-Volmer equation [77] at each light level:

$$NPQ = \frac{F_m}{F_m'} - 1$$

## Statistical assessment

To assess the effect of Fe concentration (*Control* and *+Fe*) and $pCO_2$ (*190* and *290*) on all experimental parameters among the different treatments of *P. subcurvata*, we used a two-way analysis of variance (2-way ANOVA) followed by a pairwise multiple comparison test (post hoc) using the Holm-Sidak method. All statistical analyses and the curve fittings were performed using the program SigmaPlot (Version 13.0 from Systat Software, Inc., San Jose California USA, www.systatsoftware.com). Statistical significance was defined when $p < 0.05$.

## Results

### Trace metal and carbonate chemistry

The total dFe concentrations of the different culture medium showed a significant difference between the *+Fe* and the *Control* treatments (2-way ANOVA: $p < 0.001$, Table 1), with the *+Fe* treatments having higher dFe concentrations than the *Control* treatments. The parameters of the carbonate system are given in Table 1. TA remained constant in all culture media and incubation bottles. As expected, increasing $pCO_2$ significantly enhanced the DIC concentration in all culture media and incubation bottles (2-way ANOVA: $p < 0.001$; post hoc *+Fe*: $p < 0.001$; post hoc *Control*: $p = 0.005$). While Fe availability did not alter DIC of the different culture media bottles, a significant Fe effect was found for the *P. subcurvata* incubations, but only for the *190* treatments (post hoc: $p < 0.04$). The interaction of $CO_2$ and Fe also led to significant effects in DIC of the *P. subcurvata* incubations (2-way ANOVA: $p < 0.02$). As expected, the $pCO_2$ and DIC in all of the 290 treatments were significantly higher than in the 190 treatments (2-way ANOVA; $p<0.001$; Table 1). Biologically driven changes to the carbonate chemistry were ruled out since TA, DIC, and $pCO_2$ values did not differ between the abiotic culture medium and the corresponding *P. subcurvata* incubations for each treatment at the end of the experiment (Table 1).

### Growth and elemental composition

The growth rates of *P. subcurvata* were unaffected by Fe deficiency and changes in $pCO_2$ (Fig 1A). Similarly, cell volumes remained constant across all treatments (Table 2).

**Table 1. Total dissolved iron (dFe) concentrations and carbonate chemistry determined at the end of the experiment in the culture medium (filtered seawater without cells) and the *P. subcurvata* incubations of the four treatments (*+Fe 190, Control 190, +Fe 290* and *Control 290*).** The $pCO_2$ was calculated from measured dissolved inorganic carbon (DIC) and total alkalinity (TA). For the culture medium, dFe, TA, DIC and $pCO_2$ values represent the range of duplicate abiotic controls. TA, DIC and $pCO_2$ values of the *P. subcurvata* incubations represent the means ± SD (n = 3). Differences between the individual treatments of the *P. subcurvata* incubations were determined with post hoc tests, where significant statistical ($p < 0.05$) differences are denoted by different letters.

| Parameter | Culture medium | | | |
|---|---|---|---|---|
| | **190** | | **290** | |
| | **+Fe** | **Control** | **+Fe** | **Control** |
| dFe (nmol $L^{-1}$) | 2.92–3.10 | 0.94–1.07 | 1.36–1.41 | 0.37–0.50 |
| TA (µmol $kg^{-1}$) | 2308–2318 | 2304–2319 | 2304–2323 | 2302–2311 |
| DIC (µmol $kg^{-1}$) | 2077–2101 | 2058–2077 | 2125–2131 | 2131–2132 |
| $pCO_2$ (µatm) | 208–249 | 201–208 | 269–308 | 296–309 |
| | ***P. subcurvata* incubations** | | | |
| Parameter | **190** | | **290** | |
| | **+Fe** | **Control** | **+Fe** | **Control** |
| dFe (nmol $L^{-1}$) | - | - | - | - |
| TA (µmol $kg^{-1}$) | 2317 ± 11 [a] | 2326 ± 9 [a] | 2327 ± 13 [a] | 2320 ± 9 [a] |
| DIC (µmol $kg^{-1}$) | 2046 ± 17 [a] | 2071 ± 11 [b] | 2138 ± 14 [c] | 2118 ± 2 [c] |
| $pCO_2$ (µatm) | 181 ± 15 [a] | 202 ± 24 [a] | 287 ± 31 [b] | 283 ± 29 [b] |

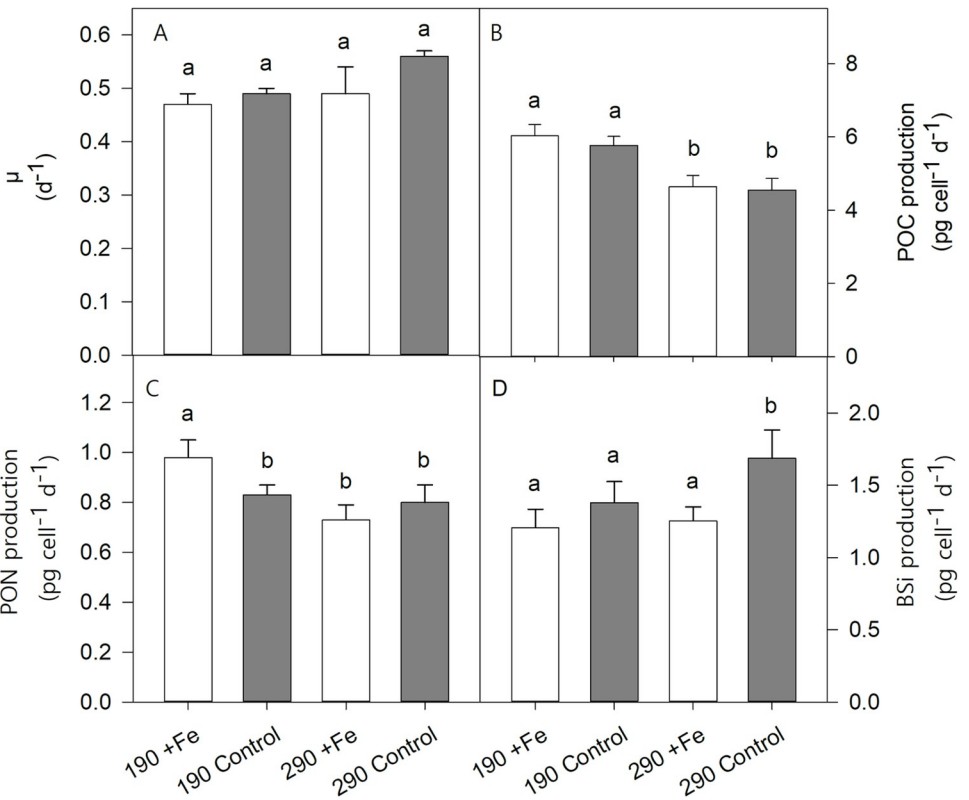

**Fig 1.** Effects of Fe reduction (*+Fe* vs *Control*) and $pCO_2$ increase (*190* vs *290*) on (A) growth rate ($\mu$), (B) POC production, (C) PON production and (D) BSi production in the four treatments of *P. subcurvata* (*+Fe 190*, *Control 190*, *+Fe 290* and *Control 290*) at the end of the experiment. The values represent the means ± SD (n = 3). Different letters indicate significant differences between treatments ($p < 0.05$).

Cellular POC quotas (Table 2) and POC production rates (Fig 1B) in both $pCO_2$ treatments were not affected by Fe deficiency. On the other hand, the increase of $CO_2$ concentration resulted in a 20–30% decrease of cellular POC quotas (2-way ANOVA: $p < 0.001$; Table 2) and POC production (2-way ANOVA: $p < 0.001$; Fig 1B) in both *Control* and *+Fe* treatments.

At *190*, lowered Fe concentration led to a decrease of cellular PON concentrations by 19% (post hoc: $p < 0.03$), while no Fe effect was observed at *290*. In response to increasing $pCO_2$, the cellular PON concentration was strongly reduced (2-way ANOVA: $p = 0.005$; Table 2) in the *+Fe* (post hoc: $p < 0.004$), but not in the *Control* treatments (Table 2). The PON production (Fig 1C) followed the same pattern as cellular PON quotas, showing a significant decrease of 15% with reduced Fe availability in the *190* treatments (post hoc: $p < 0.03$), while remaining constant in the *290* treatments. With increasing $pCO_2$, a loss of 26% in PON production in the *+Fe* (post hoc: $p < 0.02$), but not in the *Control* treatments was observed, resulting from an interactive effect of Fe and $CO_2$ availability (2-way ANOVA: $p < 0.02$; Fig 1C).

Molar C:N ratios ranged between 6.9 ± 0.1 and 8.1 ± 0.2 mol $mol^{-1}$. Fe deficiency led to a 13% increase in the C:N ratio in the *190* treatments (post hoc: $p < 0.04$), while no such Fe effect was observed in the *290* treatments. Furthermore, the increase of $CO_2$ concentration resulted in a decline of C:N by 15% in the *Control* (post hoc: $p < 0.02$), but not in the *+Fe* treatments. The interaction of Fe and $CO_2$ altered C:N ratios significantly (2-way ANOVA: $p < 0.03$; Table 2).

**Table 2. Volume and elemental composition determined at the end of the experiment in the four treatments of _P. subcurvata_ (+_Fe 190_, _Control 190_, +_Fe 290_ and _Control 290_).** The values represent the means ± SD (n = 3). Different letters indicate significant differences between treatments (p < 0.05).

| Parameter | _P. subcurvata_ incubations | | | |
|---|---|---|---|---|
| | 190 | | 290 | |
| | +Fe | Control | +Fe | Control |
| Volume ($\mu m^3$) | 31 ± 11 [a] | 34 ± 13 [a] | 34 ± 16 [a] | 32 ± 18 [a] |
| POC (pg C cell$^{-1}$) | 12.8 ± 0.9 [a] | 11.7 ± 0.8 [a] | 9.4 ± 0.9 [b] | 8.2 ± 0.6 [b] |
| PON (pg N cell$^{-1}$) | 2.1 ± 0.1 [b] | 1.7 ± 0.1 [a] | 1.5 ± 0.2 [a] | 1.5 ± 0.2 [a] |
| C:N (mol mol$^{-1}$) | 7.2 ± 0.6 [a] | 8.1 ± 0.2 [b] | 7.4 ± 0.4 [a] | 6.9 ± 0.1 [a] |
| BSi (pg Si cell$^{-1}$) | 2.6 ± 0.2 [a] | 2.8 ± 0.4 [a] | 2.6 ± 0.2 [a] | 3.1 ± 0.5 [a] |

Neither low Fe concentrations nor increased $pCO_2$ changed the cellular BSi quota (Table 2). However, as a result of Fe deficiency the BSi production in _290_ significantly increased by 35% (2-way ANOVA: p = 0.007; post hoc: p = 0.006; Fig 1D), but not in _190_. A response to higher $pCO_2$ resulted in higher BSi production only in the _Control_ treatments (post hoc: p < 0.04).

## Pigment composition

All quantified pigments, except for Chl $c_2$, were significantly affected by Fe deficiency in either the _190_ or the _290_ treatments (2-way ANOVA: Chl _a_ p < 0.001; Fuco p < 0.02; Dd p < 0.02; Dt p < 0.02; Fig 2A and Table 3). At _190_, reduced Fe availability resulted in a decrease of Chl _a_

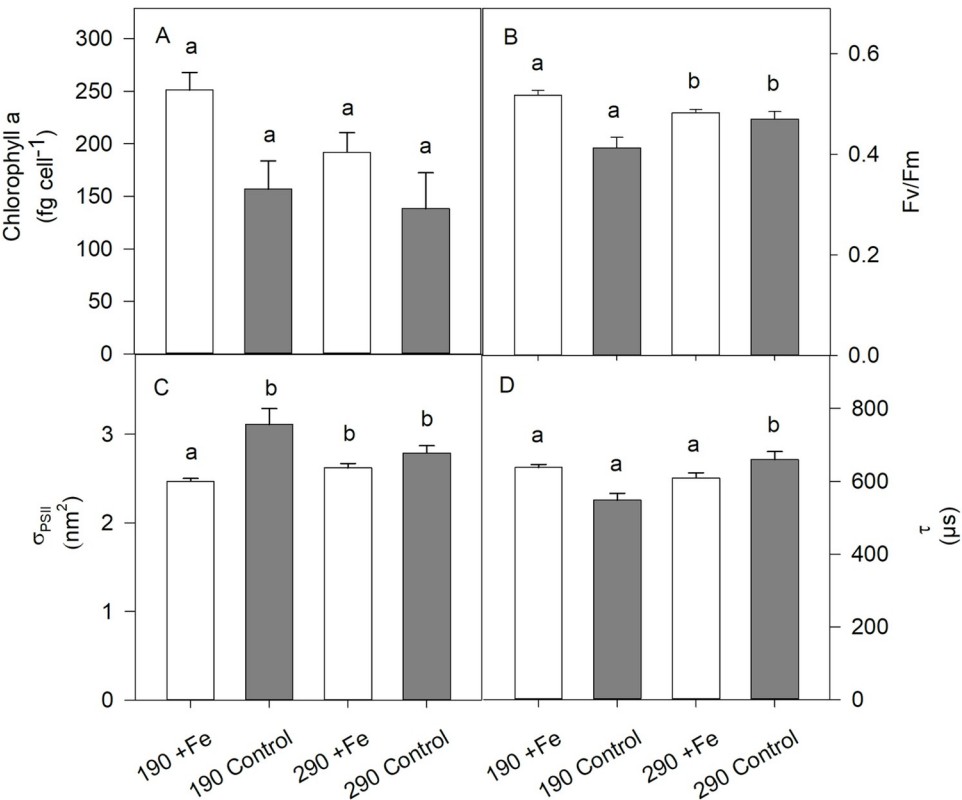

**Fig 2.** Effects of Fe deficiency and $pCO_2$ increase on (A) chlorophyll _a_ (Chl _a_), (B) photosynthetic yields ($F_v/F_m$), (C) functional absorption cross sections ($\sigma_{PSII}$) and (D) time constants ($\tau$) in the four treatments of _P. subcurvata_ (+_Fe 190_, _Control 190_, +_Fe 290_ and _Control 290_) at the end of the experiment. The values represent the means ± SD (n = 3). Different letters indicate significant differences between treatments (p < 0.05).

**Table 3. Pigment concentrations determined at the end of the experiment in the four treatments of *P. subcurvata* (+*Fe 190*, *Control 190*, +*Fe 290* and *Control 290*).** The values represent the means ± SD (n = 3). Different letters indicate significant differences between treatments.

| Parameter | *P. subcurvata* incubations | | | |
|---|---|---|---|---|
| | 190 | | 290 | |
| | +Fe | Control | +Fe | Control |
| Chlorophyll $c_2$ (fg cell$^{-1}$) | 28.9 ± 6.9 [a] | 19.4 ± 5.7 [a] | 22.3 ± 5.1 [a] | 19.1 ± 5.0 [a] |
| Fucoxanthin (fg cell$^{-1}$) | 140 ± 10 [a] | 93 ± 24 [b] | 110 ± 16 [a] | 86 ± 24 [a,b] |
| Diadinoxanthin (fg cell$^{-1}$) | 28.3 ± 3.8 [a] | 19.4 ± 4.5 [b] | 24.1 ± 4.3 [a] | 17.7 ± 3.9 [a,b] |
| Diatoxanthin (fg cell$^{-1}$) | 1.27 ± 0.24 [a] | 1.08 ± 0.12 [a,b] | 1.47 ± 0.44 [a] | 0.64 ± 0.21 [b] |
| Chl *a*:C (mol mol$^{-1}$) | 0.21 ± 0.03 [a] | 0.14 ± 0.02 [b] | 0.22 ± 0.04 [a] | 0.16 ± 0.03 [b] |

by 37% (post hoc: p = 0.002), of Fuco by 34% (post hoc: p < 0.02) and of Dd by 29% (post hoc: p = 0.03), while Dt was not affected. At *290*, the reduction of Fe significantly reduced the Chl *a* concentration by 23% (post hoc: p = 0.03) and Dt by 60% (post hoc: p = 0.007), whereas Fuco and Dd remained constant. In response to elevated $pCO_2$, cellular Chl *a* quotas of *P. subcurvata* were significantly reduced in the +*Fe* (251 ± 17 to 192 ± 19 fg cell$^{-1}$ for *190* and *290*, respectively; 2-way ANOVA: p < 0.03; post hoc: p = 0.02; Fig 2A), while this trend was absent in the *Control*. No other pigments (Fuco, Chl $c_2$, Dd or Dt) responded to changes in the $pCO_2$.

The Chl *a*:C ratio in *P. subcurvata* was significantly affected by Fe deficiency (2-way ANOVA: p = 0.005; Table 3) leading to a decrease of 33% (post hoc: p < 0.03) and 27% (post hoc: p < 0.04) in the *190* and *290* treatments, respectively. Conversely, increased $pCO_2$ had no effect on the Chl *a*:C ratio.

## Maximum quantum yield and changes to PSII

The photosynthetic yield of *P. subcurvata* ($F_v/F_m$) showed a significant Fe effect (2-way ANOVA: p < 0.001; Fig 2B). At *190*, $F_v/F_m$ decreased significantly by 21% in response to Fe deficiency (from 0.52 ± 0.01 to 0.41 ± 0.02 in the +*Fe* and *Control*, respectively, post hoc: p < 0.001), while no Fe effect was observed in *290*. Interestingly, $CO_2$ enhancement differently affected the photosynthetic yield of the two Fe treatments. While increasing $pCO_2$ enhanced the $F_v/F_m$ in the *Control* treatment by 15% (from 0.41 ± 0.02 to 0.47 ± 0.01, post hoc: p = 0.005), it reduced $F_v/F_m$ in the +*Fe* treatments by 8% (from 0.52 ± 0.01 to 0.48 ± 0.01, post hoc: p < 0.04). Hence, there was a significant interactive effect of $CO_2$ and Fe availability on $F_v/F_m$ (2-way ANOVA: p = 0.002; Fig 2B).

The connectivity (P) was significantly affected by Fe deficiency (2-way ANOVA: p = 0.002; Table 4), with the *Control* treatment having an 11% smaller energy transfer between PSII units

**Table 4. Connectivity (P), cellular concentration of functional PSII reaction centers (RCII), light utilization efficiency at low irradiance (α), maximum cellular electron transport rate (cETR$_{max}$) and minimum saturating irradiance ($I_k$) of *P. subcurvata* in the four treatments (+*Fe 190*, *Control 190*, +*Fe 290* and *Control 290*) at the end of the experiment.** The values represent the means ± SD (n = 3). Different letters indicate significant differences between treatments (p < 0.05).

| Parameter | *P. subcurvata* incubations | | | |
|---|---|---|---|---|
| | 190 | | 290 | |
| | +Fe | Control | +Fe | Control |
| P (rel. unit) | 0.44 ± 0.01 [a] | 0.39 ± 0.02 [b] | 0.43 ± 0.01 [a] | 0.40 ± 0.01 [a,b] |
| RCII (zmol cell$^{-1}$) | 515 ± 58 [a] | 525 ± 42 [a] | 370 ± 38 [b] | 519 ± 47 [a] |
| α (amol e$^-$ cell$^{-1}$ s$^{-1}$/ μmol photons m$^{-2}$ s$^{-1}$) | 0.75 ± 0.13 [a] | 0.97 ± 0.14 [b] | 0.58 ± 0.08 [a] | 0.82 ± 0.07 [b] |
| cETR$_{max}$ (amol e$^-$ cell$^{-1}$ s$^{-1}$) | 119 ± 21 [a] | 165 ± 26 [b] | 85 ± 5 [a] | 139 ± 19 [b] |
| $I_k$ (μmol photons m$^{-2}$ s$^{-1}$) | 155 ± 9 [a] | 171 ± 11 [a,b] | 143 ± 15 [a] | 169 ± 9 [b] |

than the +*Fe* at *190* (post hoc: p = 0.002). In the *290* treatments, a similar, however, not significant, decreasing trend was seen. In contrast, no response of P to increased $CO_2$ was observed.

The functional absorption cross section of PSII ($\sigma_{PSII}$) showed a significant effect to Fe deficiency (2-way ANOVA: p < 0.001; Fig 2C). While $\sigma_{PSII}$ increased by 26% with reduced Fe availability in *190* (from 2.47 ± 0.03 to 3.11 ± 0.18 $nm^{-2}$, respectively, post hoc: p < 0.001), this Fe effect was not seen in the *290* treatments. Furthermore, only in the *Control* treatments $\sigma_{PSII}$ was reduced by 10% from 3.11 ± 0.21 to 2.79 ± 0.09 $nm^{-2}$ between *190* and *290*, respectively (post hoc: p = 0.01). Moreover, there was a synergistic effect between Fe and $CO_2$ on $\sigma_{PSII}$ (2-way ANOVA: p = 0.009; Fig 2C).

The cellular concentration of functional PSII reaction centers (RCII) was significantly altered by Fe deficiency (2-way ANOVA: p < 0.04; Table 4). This effect was only seen in *290*, where RCII increased by 29% (post hoc: p < 0.02). Increasing $CO_2$ significantly reduced the RCII concentration (2-way ANOVA: p < 0.05), but only in the +*Fe* treatments (post hoc: p < 0.02).

Fe deficiency differently influenced the time constant for electron transport at the acceptor of PSII ($\tau$) in the two $CO_2$ treatments. While lower Fe concentration reduced $\tau$ when grown at 190 µatm $pCO_2$ (post hoc: p < 0.001), it was enhanced at 290 µatm $pCO_2$ (post hoc: p = 0.006; Fig 2D). The effect of increased $CO_2$ on $\tau$ was significant (2-way ANOVA: p < 0.004). In the *Control* treatments, $\tau$ increased from 548 ± 21 to 659 ± 23 µs from 190 to 290 µatm $pCO_2$ (post hoc: p < 0.001) while it remained constant in the +*Fe* treatments. Hence, there was a strong interactive effect of Fe and $CO_2$ on $\tau$ apparent (2-way ANOVA: p < 0.001).

## PE-curve

The cellular electron transport rates (cETR) of all treatments followed the shape of a typical PE-curve (Fig 3A). The light utilization efficiency of *P. subcurvata* at low irradiance ($\alpha$) was significantly affected by Fe deficiency (2-way ANOVA: p = 0.005; Table 4), with $\alpha$ increasing by 29% at *190* (post hoc: p < 0.04) and by 41% at *290* (post hoc: p < 0.02). A $CO_2$ effect was also observed (2-way ANOVA: p = 0.02), where increased $CO_2$ reduced $\alpha$, but due to large uncertainties, the individual post hoc tests of the +*Fe* and *Control* treatments were not significant. In response to Fe deficiency, $cETR_{max}$ (Table 4 and Fig 3A) was significantly enhanced (2-way ANOVA: p < 0.006) by 39% at *190* and by 64% at *290* (both post hoc: p < 0.03). The increase in $CO_2$, however, did not lead to significant changes in $cETR_{max}$. The minimum saturating irradiance ($I_k$) displayed a significant Fe effect (2-way ANOVA: p < 0.02; Table 4), where $I_k$ increased by 10% in the *290* treatments (post hoc: p < 0.04). Although not significant (p>0.05), in the *190* treatments a similar trend was observed. $I_k$ remained unchanged by increasing $CO_2$ irrespective of Fe availability.

The non-photochemical quenching of all treatments was similarly low at low irradiances (Fig 3B). Exposed to irradiances higher than 350 µmol photons $m^{-2}$ $s^{-1}$, the NPQ in *P. subcurvata* increased nearly linearly and then leveled off between ~1.5 and 2.5 for all treatments. No Fe or $CO_2$ effect on NPQ was observed in any treatment.

## Discussion

The 'Iron Hypothesis' suggests that the fertilization of the SO by increased dust deposition in glacial times promoted growth and productivity of phytoplankton. The biological pump in the SO was thus hypothesized to have reduced atmospheric $pCO_2$. In this study, we assessed the ecophysiological response of *P. subcurvata* simulating glacial and interglacial climate scenarios in terms of changes in Fe and $CO_2$ availability. It is important to note that while we manipulated two of the main environmental parameters, Fe concentrations and $pCO_2$, other

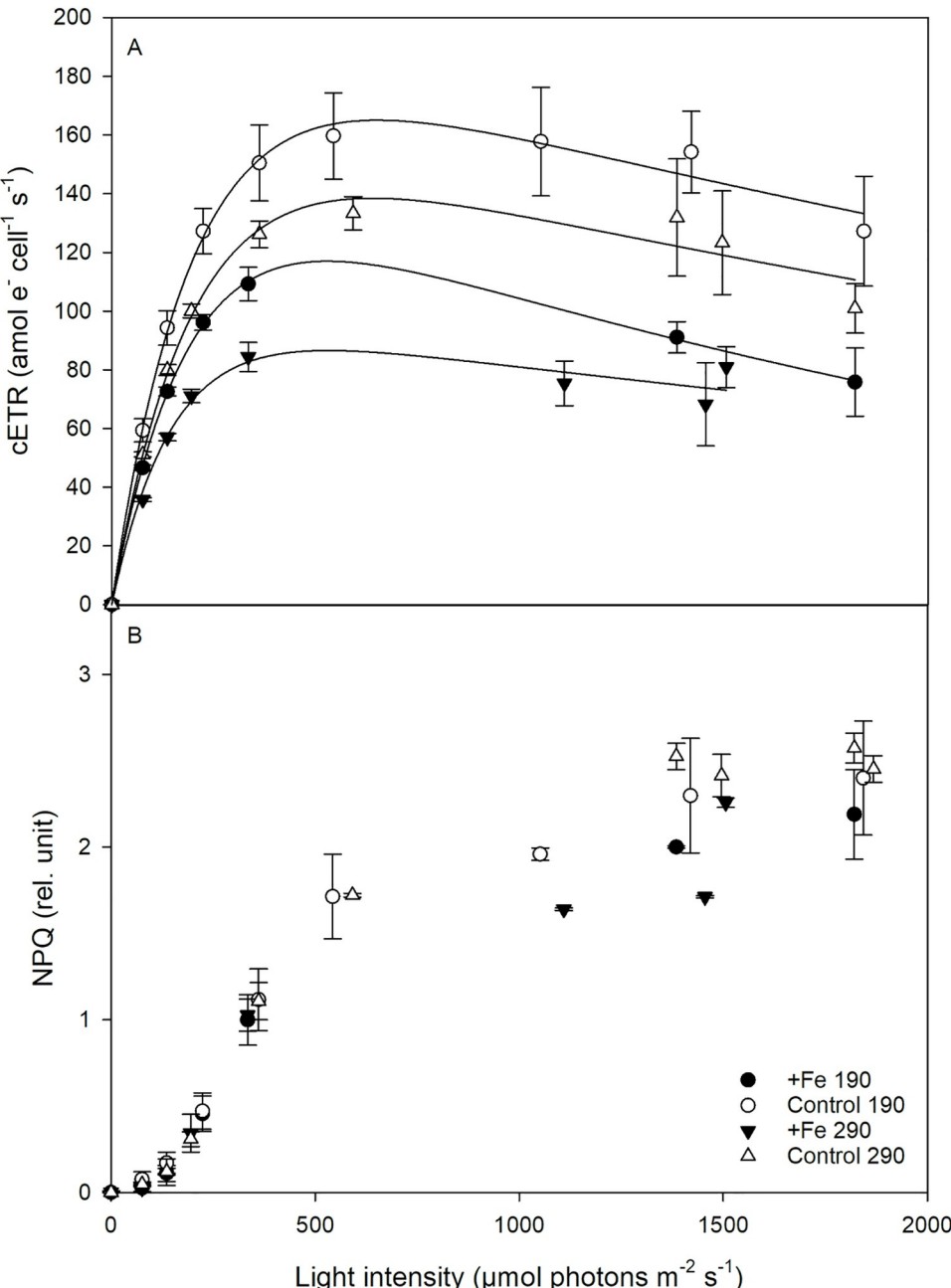

**Fig 3.** Effects of Fe deficiency and $CO_2$ increase on (A) cellular electron transport rates (cETR) and on (B) non-photochemical quenching (NPQ) in the four treatments with *P. subcurvata* (*+Fe 190*, *Control 190*, *+Fe 290* and *Control 290*) at the end of the experiment. The values represent the means ± SD (n = 3).

parameters (macronutrient concentrations, temperature i.e.) likely also differed between the glacial and interglacial ocean. For this study, however, the focus was on the interactive effects of Fe and $pCO_2$.

## Glacial conditions favored POC production by *P. subcurvata*

Between 190 and 290 μatm $pCO_2$, no change in growth rate was observed in the *+Fe* treatments of *P. subcurvata* (Fig 1A). Previous laboratory studies with cultures of the same *P.*

*subcurvata* strain also reported no changes in growth rate between 180 and 390 μatm $pCO_2$ [51]. Similarly, growth remained unaffected in the temperate *Pseudo-nitzschia pseudodelicatissima* between 200 and 380 μatm $pCO_2$ [78], in *T. pseudonana*, *T. rotula*, and *T. oceanica* from 230 to 350 ppm [79] and in *Proboscia alata* from 135 to 200 μatm $pCO_2$ [47]. Additionally, growth rates, pigment contents, photosynthesis and photophysiology of the Antarctic diatom *Chaetoceros brevis* did not change between 190 and 750 ppm [80]. Differently, however, is the study by [46], which reported a stimulation of the growth rate of another *P. subcurvata* strain from 100 up to 450 μatm $pCO_2$. Also, growth of the temperate *Pseudo-nitzschia multiseries* was enhanced between 220 and 400 ppm $pCO_2$ [48]. It appears therefore that species- and strain-specific differences in the $CO_2$-dependence of growth among *Pseudo-nitzschia* exist.

The similar growth rates at both $pCO_2$ levels and Fe availabilities maintained by *P. subcurvata* in our experiment (Fig 1A) suggest the operation of carbon concentrating mechanisms (CCMs), which efficiently avoided $CO_2$ limitation. This can also be inferred from [50], where *Pseudo-nitzschia* was the most abundant species within a natural Southern Ocean phytoplankton assemblage under both Fe-enriched and Fe-deplete conditions at 180 and 390 μatm. Previous studies showed that Antarctic phytoplankton species such as *P. subcurvata* operate very efficient CCMs, which are constitutively expressed irrespective of $CO_2$ availability [49, 51, 81]. In addition to highest uptake rates of C and macronutrients, the temperate diatom *P. pseudodelicatissima* exhibited a high Fe uptake affinity at 170 ppm [78]. The latter findings indicate that *Pseudo-nitzschia* species can cope well with low $CO_2$ conditions, enabling them to maintain high growth even under low $CO_2$ conditions, as can be also seen here in *P. subcurvata* (Fig 1A).

In this experiment, $F_v/F_m$ was highest in the *+Fe 190* treatment (Fig 2B), indicating that *P. subcurvata* possessed highest photochemical fitness under simulated glacial conditions. With increasing $pCO_2$, however, $F_v/F_m$ declined in the diatom, but only when Fe was added (Fig 2B). Such a negative $CO_2$ effect in Fe-enriched conditions was also observed in the Chl *a* content (Fig 2A) and the number of functional RCII (Table 4). Indeed, *P. subcurvata* cells grown in the *+Fe 290* treatment had a lower Chl *a* content compared to ones in the *+Fe 190* treatment (Fig 2A), although the Chl *a*:C ratios were similar.

Moreover, cellular BSi quotas and production remained constant with increasing $pCO_2$ in the *+Fe* treatments (Fig 1D, Table 2) while a decline in POC and PON quotas as well as in POC and PON production rates (Fig 1B and 1C and Table 2) was found. Reducing both POC and PON quotas, *P. subcurvata* was able to maintain a constant C:N ratio (Table 2) in response to increasing $pCO_2$ under Fe-enriched conditions. Considering, however, that cETRs remained similar between *190* and *290* (Fig 3A, Table 4), a reduction in POC and PON contents indicates that the contribution of linear electron transport was reduced while cycling of electron via alternative pathways was required to avoid excess light energy. These physiological characteristics resemble those observed in various field incubation experiments under ocean acidification conditions and indicate that *P. subcurvata* struggles when exposed to high $pCO_2$ levels [24, 49, 50]. Overall, we can conclude that glacial conditions simulated by a low $pCO_2$ of 190 μatm together with Fe enrichment was neither limiting growth nor POC production of *P. subcurvata*. On the contrary, these conditions were beneficial for biomass production and photochemical fitness of the diatom.

## *P. subcurvata* adjusted its physiological machinery to cope with low Fe supply

Contrary to other studies, we did not observe a decrease in cell volume of *P. subcurvata* grown with decreasing Fe availability (Table 2) [42, 82]. This may have been masked by the fact that

the *P. subcurvata* strain used in our experiment was acclimated to low Fe conditions for a long time. Indeed, it exhibited large and elongated cells compared to the much shorter cells of the stock culture grown in the Fe-rich F$_2$ medium (12 μM Fe), thus increasing their surface area-to-volume ratio. Furthermore, this strain was isolated from open ocean waters in the Atlantic sector of the SO. It is well known that oceanic diatoms acclimate to Fe limitation by increasing their surface area-to-volume ratio in order to maximize the number of transporter sites and nutrient uptake kinetics [83, 84].

Many studies reported a decrease in growth rate with decreasing Fe availability [8, 39, 40, 42, 44, 85–89]. Nonetheless, some of them also observed that particular oceanic diatoms grew at comparable rates under high and low Fe conditions [8, 86], as they have evolved acclimation strategies to reduce their Fe requirement. In our experiment, the growth rate of the oceanic *P. subcurvata* also displayed no difference between *+Fe* and *Control* conditions at the two pCO$_2$ levels tested (Fig 1A) [42]. Suggested that the response of physiological and biochemical parameters to Fe reduction precedes changes in growth rate. This may explain why we did not see a decrease in growth rate here, despite observing typical responses to Fe-limiting conditions as substantial reductions in photochemical quantum efficiency (Fig 2B), connectivity (Table 4) and Chl *a* content (Fig 2A) accompanied by large functional absorption cross sections (Fig 2C) [9, 38, 40, 41, 85, 87, 88, 90, 91].

Under Fe deficiency, lowered F$_v$/F$_m$ values indicate that the excitation energy was less efficiently transferred in the antennae, due to damaged and altered parts of the photosynthetic apparatus [41]. A decrease in F$_v$/F$_m$ was commonly observed in cells grown in Fe-poor environments [9, 40, 41, 44, 87, 88] and, as expected, we observed this trend in the *Control 190* treatment of *P. subcurvata* (Fig 2B). In line with the tested *P. subcurvata* here, oceanic *Pseudo-nitzschia* species usually decouple F$_v$/F$_m$ and growth rate, reducing the former while maintaining the latter [8]. This decoupling was suggested to be due to either a low energy requirement of the diatom, or a compensating mechanism that generates reducing power, thus supporting rapid growth [8].

The decrease in F$_v$/F$_m$ (Fig 2B) and lowered connectivity (P, Table 4) at low pCO$_2$ in the low Fe *P. subcurvata* cells indicate that the transfer of excitation energy to the reaction centers was compromised [9]. Because Fe deficiency affects the synthesis and thus cellular content of Chl *a*, as seen in our data (Fig 2A), light harvesting may become more difficult for the cell. While [85] held lowered pigment concentration during Fe starvation responsible for a decline in photosynthesis, we did not observe reduced POC production rates (Fig 1B). Rather *P. subcurvata* compensated for a low Chl *a* content by increasing the functional absorption cross section of PSII ($\sigma_{PSII}$), which is a measure of the target area of the light harvesting antenna (Fig 2C). In response to Fe deficiency this strategy can reduce the Fe demand and keep up the same capacity of the cell to absorb light [92]. Our results agree with literature showing an increase in $\sigma_{PSII}$ with Fe reduction [9, 38, 40, 41, 75, 87–89].

These photophysiological adjustments, however, did not prevent changes in light absorption completely, as shown by the strongly impacted light use capacities of Fe-limited *P. subcurvata* (Table 4). Higher α values were found under Fe deficiency for both *190* and *290* treatments, indicating that cells were able to respond better to lower irradiances than Fe-replete cells. Surprisingly, this effect was not always observed for I$_k$ values of *P. subcurvata*, which remained similar at *190* and was slightly higher at 290 (Table 4). Thus, while Fe deficiency at 290 resulted in a more efficient light utilization at lower irradiances (higher α), the cells required more light (higher I$_k$) in order to cover their photosynthetic requirement [40]. In other studies, I$_k$ either decreased [40, 93] or remained unchanged [44, 85] under Fe reduction.

Even though POC-fixation remained constant under Fe deficiency (Fig 1B), $cETR_{max}$ (Fig 3A, Table 4) and RCII concentration (only seen at 190 µatm $pCO_2$) were enhanced (Table 4), indicating similar linear electron transport, but also cycling of electrons into alternative pathways such as cyclic electron flow within PSII [94] or Mehler reaction [95]. Considering, however, that the latter pathways are Fe-expensive, other pathways such as activity of a putative plastid plastoquinol terminal oxidase (PTOX) seem more plausible [96]. In support for this, [9] also observed constant C assimilation, but enhanced electron transport with Fe limitation in open ocean phytoplankton. Furthermore, a quicker turnover time at the acceptor side of PSII (τ) was found at 190 µatm $pCO_2$ in the Fe deficient *P. subcurvata* cells (Fig 2D), supporting PTOX activity, as previously observed for the Fe-limited Antarctic diatom *Chaetoceros debilis* [44]. Interestingly, this was not reflected in higher NPQ activities (Fig 3B).

At low $pCO_2$, BSi quotas and production rates of *P. subcurvata* remained unaltered in response to Fe deficiency (Fig 1D, Table 2), as previously observed in *Chaetoceros debilis* [44], *Corethron pennatum* [97] and *Chaetoceros dichaeta* [98]. Considering the importance of Fe in C and N assimilation pathways, many studies reported a decrease in C and N under Fe deficiency [40, 42]. In [99], the C quota per cell volume ranged between 0.02 and 0.03 pg $µm^{-3}$ and was similar between Fe-replete and Fe-deficient treatments in the oceanic *Pseudo-nitzschia fraudulenta*, *P. heimii*, *P. inflatula* and *P. turgidula*, as well as in the coastal species *P. multiseries* and *P. pseudodelicatissima*. This matches with our results for the two tested $pCO_2$ levels (POC per cell volume at *190 +Fe*: 0.041±0.002 pg $µm^{-3}$, *Control*: 0.035±0.005 pg $µm^{-3}$ and at *290 +Fe*: 0.029±0.005 pg $µm^{-3}$, *Control*: 0.027±0.003 pg $µm^{-3}$). The C:N ratio of diatoms was reported to increase [78], decrease [100] or remain unchanged [42, 44] with reduced Fe availability. We observed an increase in the C:N ratio in response to Fe deficiency at 190 µatm $pCO_2$ (Table 2). In this case, POC quotas remained constant, whereas PON cell quotas decreased with Fe deficiency (Table 2). Literature showed that Fe limitation can affect the supply of 'new nitrogen' to the cell as Fe is needed in some N-rich enzymes [101, 102]. [75] observed less abundant transcripts for nitrite reductase under Fe limiting conditions in *Phaeocystis antarctica*. Considering this, our reduced PON-fixation in *P. subcurvata* under low Fe conditions in conjunction with low $pCO_2$ could be coupled to a protein recycling process to avoid N-limitation [39, 75, 103].

We can conclude that Fe deficiency results in a less efficient transfer of excitation energy in *P. subcurvata*, allowing it to reduce its Fe demand. In order to keep up the same POC production, *P. subcurvata* needed to rely on alternative electron pathways such as cyclic electron flow as well as PTOX activity to prevent over-excitation.

## Increased $pCO_2$ weakened the effects of low Fe supply, but did not promote biomass build up

Previous experiments with *Pseudo-nitzschia* demonstrated on the one hand, that the cell volume of *P. pseudodelicatissima* increased significantly as $pCO_2$ decreased, while, on the other hand, cell volume was found to decrease with decreasing Fe availability [42, 82]. In our experiments, the cell volume of *P. subcurvata* did not decrease with reduced Fe availability and increased $pCO_2$ (Table 2), potentially due to a counteracting effect of both factors together. Moreover, $F_v/F_m$ decreased in response to Fe reduction at *190* (Fig 2B), while such Fe-dependent decrease in $F_v/F_m$ was not observed at *290*. This indicates that increasing $pCO_2$ had a positive effect on the maximum photochemical efficiency of low Fe *P. subcurvata* cells. A similar effect by high $CO_2$ concentration was also found for $σ_{PSII}$ in Fe-deplete cells, being much smaller (Fig 2C). Apparently, these positive $CO_2$ effects weakened the strong Fe reduction effects previously observed at *190*. Such positive response did, however, not translate into

more efficient energy transfer from photochemistry to biomass production. In fact, re-oxidation of the primary electron acceptor $Q_a$ of low Fe cells was strongly compromised at *290* (Fig 2D). This was associated with reduced POC fixation and enhanced cETRs at *290* (Fig 3A, Table 4), and as a consequence, alternative electron acceptors were required. Due to a synergetic effect of reduced Fe availability and increased $pCO_2$, in our experiment we observed the highest BSi production in low Fe high $pCO_2$ conditions (Fig 1D). This increase in BSi production with reduced Fe concentrations at *290* hints towards stronger silicification and the production of thicker shells by *P. subcurvata* [104–106].

## Conclusion: Glacial vs. interglacial

In our study, in a simulated Fe-fertilized glacial ocean (+*Fe 190*), *P. subcurvata* displayed similar growth rates as in interglacial ocean conditions (*Control 290*), despite lower Fe availability, hinting towards an efficient acclimation strategy to reduce the Fe requirement. Under glacial conditions, electrons were more efficiently channeled, leading to higher cellular POC and PON concentrations and production rates. In comparison, the interglacial conditions with higher $pCO_2$ and reduced Fe availability resulted in reduced POC buildup of the diatom. Thus, we observed that both higher Fe availability and lower $CO_2$ concentration as in the glacial ocean, promoted POC production by *P. subcurvata*. Assuming that *P. subcurvata* dominated phytoplankton blooms in the SO during glacial and interglacial times, we can conclude that *P. subcurvata* contributed more to primary production in the glacial than interglacial ocean. The higher POC production rates by the diatom under glacial conditions facilitated higher $CO_2$ uptake from the atmosphere and potentially higher C export. This matches the 'Iron Hypothesis' of [1], which states that in the last glacial maximum higher Fe input from dust fertilized the SO, thus stimulating higher primary production and reducing thereby the atmospheric $CO_2$ concentration. On the other hand, however, the thicker shells of *P. subcurvata* under the simulated interglacial conditions hint towards reduced grazing and thus its higher contribution to C export [107]. Biogeochemical cycles changed in the past and will change in response to future global climate change. Thus, understanding the dynamic interactions of the ocean's biogeochemistry and phytoplankton is important in order to better simulate past and future climatic scenarios.

## Supporting information

**S1 File. Cellular trace metal quotas.** Trace metal (TM) quotas without oxalate (Total TM content) and with oxalate wash (Intracellular TM content) determined at the end of the experiment in the four treatments of *P. subcurvata* (+*Fe 190*, *Control 190*, +*Fe 290* and *Control 290*). The values represent the means ± SD (n = 3). Different letters indicate significant ($p < 0.05$) differences between treatments.
(DOCX)

## Acknowledgments

We thank (in alphabetical order) T. Brenneis, C. Völkner and D. Wilhelms-Dick for laboratory assistance and for analyzing the samples. Thanks also to K. Bischof and B. Meyer-Schlosser for the pigment analysis.

## Author Contributions

**Conceptualization:** Anna Pagnone, Scarlett Trimborn.

**Data curation:** Anna Pagnone, Scarlett Trimborn.

**Formal analysis:** Anna Pagnone.

**Funding acquisition:** Scarlett Trimborn.

**Investigation:** Anna Pagnone.

**Methodology:** Anna Pagnone.

**Project administration:** Scarlett Trimborn.

**Resources:** Scarlett Trimborn.

**Supervision:** Scarlett Trimborn.

**Validation:** Anna Pagnone.

**Visualization:** Anna Pagnone, Florian Koch, Franziska Pausch.

**Writing – original draft:** Anna Pagnone, Florian Koch, Franziska Pausch, Scarlett Trimborn.

**Writing – review & editing:** Anna Pagnone, Florian Koch, Franziska Pausch, Scarlett Trimborn.

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
