## [Decision Letter · Decision Letter 0]

30 Sep 2021

PONE-D-21-25243The Southern Ocean diatom Pseudo-nitzschia subcurvata flourishes better under simulated glacial than interglacial ocean conditionsPLOS ONE

Dear Dr. Koch,

Thank you for submitting your manuscript to PLOS ONE. After careful consideration by two expert reviewers, we feel that it has considerable merit but does not yet fully meet PLOS ONE’s publication criteria as it currently stands. Therefore, we invite you to submit a revised version of the manuscript that addresses the points raised during the review process.

 Reviewer 1 asks for any evidence that the study diatom dominated the Southern Ocean over glacial and inter-glacial times.  Reviewer 1 also notes a subtle issue with estimating C assimilation rates from cellular C content which should be addressed carefully.  Reviewer 1 also offers some corrections of typos and points of clarification. Reviewer 2 points out issues with the presented metric of Non-PhotoChemical Quenching and suggests alternat or parallel presentation of a different metric (NPQ  NSV) which has been validated in similar studies. Reviewer 2 calls for clarification of exactly how the fluorescence metrics were captured, and offers a correction to the discussion of IKE. I am delighted your strong work can benefit from such constructive reviews, which I hope you find useful.

We look forward to receiving your revised manuscript.

Kind regards,

Douglas A. Campbell, Ph.D.

Academic Editor

PLOS ONE

Journal Requirements:

"This work was supported by German Federal Ministry of Education and Research (BMBF) as Research for Sustainability initiative (FONA); www.fona.de through Palmod project (FKZ: 01LP1505C). We thank (in alphabetical order) T. Brenneis, C. Völkner and D. Wilhelms-Dick for laboratory assistance and for analyzing the samples. Thanks also to K. Bischof and B. Meyer-Schlosser for the pigment analysis. ST, FK and FP were funded by the Helmholtz Association (HGF Young Investigators Group EcoTrace, VH-NG-901)."

"AP was supported by German Federal Ministry of Education and Research (BMBF) as Research for Sustainability initiative (FONA); www.fona.de through Palmod project (FKZ: 01LP1505C). ST, FK and FP were funded by the Helmholtz Association (HGF Young Investigators Group EcoTrace, VH-NG-901). 

Reviewers' comments:

Reviewer's Responses to Questions

**Comments to the Author**

1. Is the manuscript technically sound, and do the data support the conclusions?

Reviewer #1: Yes

Reviewer #2: Yes

2. Has the statistical analysis been performed appropriately and rigorously? 

Reviewer #1: Yes

Reviewer #2: Yes

3. Have the authors made all data underlying the findings in their manuscript fully available?

Reviewer #1: Yes

Reviewer #2: Yes

4. Is the manuscript presented in an intelligible fashion and written in standard English?

Reviewer #1: Yes

Reviewer #2: Yes

5. Review Comments to the Author

Reviewer #1: In this well-written paper, the authors reported the ecophysiological responses of diatom Pseudo-nitzschia subcurvata to the simulated glacial and interglacial climatic conditions i.e., no Fe or Fe addition under 190 and 290 μatm pCO2 levels. They found the varied Fe and pCO2 levels had a limited effect on growth rate, but individually or interactively altered the cell compositions including pigments, POC, PON and BSi, as well as photophysiology of P. subcurvata. They conclude the combined higher Fe availability and lower pCO2 (present in glacial ocean) was beneficial for the diatom P. subcurvata, thus contributing more to primary production during glacial than interglacial times. There are some minor revisions should be considered before accepted for publication in PLoS One journal.

I think the title exaggerates the results of the experiment; because, the authors just simulated two environmental factors i.e., Fe and pCO2 levels in glacial and interglacial times, that cannot represent all the environmental factors.

In Discussion part, the authors fully explained the data in the view of physiology, some ecological perspectives should be given as they did the simulated field condition experiments.

1. Line 26, The ecological significance of “The thicker silica shells present under interglacial conditions” has been referred in the end of the Abstract; so, this sentence “which might offer better protection against grazers” should be removed. Besides, the authors did not give any descriptions about grazers throughout the manuscript.

2. Is there any geological evidence to show the Pseudo-nitzschia species dominate the Southern Oceans in glacial and inter-glacial times? I think some information should mentioned in the Introduction part.

3. One of half brackets was missing in several places of the manuscript, e.g., Lines 105-106, Line 209, Line 220, Line 239;

4. In Materials and Methods part, clarify how much volume of culture was filtrated to measure the POC, PON, BSi and pigments.

5. In Line 195-196, the authors clarified the cell densities at the end of the cultivation were 67000 to 107000 cells per ml; I’m wondering what the cell density was at the initial inoculation. Please clarify it.

6. Line 280, The light intensity of fluorometer is so high. Please check it.

7. Line 252-253, Production rates of POC, PON and BSi were calculated by multiplying the cellular quotas with the respective growth rate.

It’s OK to calculate PON and BSi production rate. But I don’t think it’s right to use this way to calculate POC production rate; because, the cultures were grown under 16:8 light:dark cycle under 100 µmol photons m-2 s-1 light intensity, that means the POC production by photosynthesis just occurs in light duration (16 h per day), while the growth rate was calculated from the cell density changes of a whole day (24 h). So, they are mismatching.

8. Line 337-338, The authors clarified that “dFe values denote two measurements”; so, it is wrong to use the standard deviation here. I suggest to replace the SD to half of range.

Reviewer #2: The following manuscript provides details on the interactive effects of Fe and pCO2 on an Antarctic diatom, Pseudo-nitzschia subcurvata, under glacial and inter-glacial conditions. Whilst there is a wealth of evidence of understanding higher pCO2 concentrations, representative of future climate change, very little is known about how this diatom copes under these pre-industrial conditions. The manuscript is very well written, with clean concise results and figures. I recommend this paper for publication, with only minor corrections which are outlined below.

Minor Corrections:

Line 248: Should the NaOH concentration not be “M”, currently it is “N”.

Line 276: Were the samples collected during the day or night part of the light cycle? This will greatly impact the effects of dark adaptation (see Schuback et al. 2021 Frontiers for more details on this).

Line 284: No text is provided to determine whether blanks were collected, and the fluorescence subsequently corrected. This is an important step for any active chlorophyll fluorescence measurements.

Line 294: Please provide the number of measurements per light level. Was any QC applied to these measurements per light level? As stated in reference 76 (Ralph and Gademann) unlike traditional PE curves, the measurements do not reach steady state. That is why previous studies have opted for either taking the mean of the last 3 measurements per light level or applying a QC to remove potential outliers.

Line 302: More details are required here to indicate how the curve fitting was done. Was this done in FastPro8 or was another program used to fit the curves.

Line 306: Reference required for the Stern-Volner equation – Bilger and Bjorkman, 1991, doi 10.1007/BF00197951.

Line 329: This statement is a little confusing, based upon the significant differences above. Are you saying that the CO2 parameters are significantly different between 190 and 290? Or are you saying they are significantly different between the Fe and control? If the latter, what impact do you think it will have on the results if the CO2 parameters are also significantly different alongside the Fe concentrations.

Line 339: Unless I have missed it, I am struggling to find where the definition of the letters is stated. This makes it difficult to understand what was significant or not.

From my understanding:

a = all the same

b = significantly different to a

c = significantly different to a and b

Perhaps you can make it a little clearer for the reader.

Table 1: dFe concentrations – do you know what may have caused such differences? Additionally, a 1 nM concentration for the 190 control would not necessarily be limiting for SO phytoplankton species based upon measured in situ concentrations.

Line 378: Do you mean “but not in 190”?

Line 416: The connectivity parameter is most often reported as the Greek letter rho, ρ.

Table 4: Units of α, should this not be amol e- cell-1 s-1 (μmol photons m-2 s-1)-1?

Line 458: This does not make sense to me, in the 190 treatment the standard deviations are 9 and 11, whereas in the 290 treatments they are 15 and 9. So how is it that the large standard deviations in the 190 prevent statistical significance?

Line 505: I wouldn't necessarily agree with this statement. The Chl:C ratios are similar for both Fe treatments at 190 and 290. As well as σPSII being similar between Fe and control at 290. These are better indicators of potential light absorption for photosynthesis.

Line 540: Missing some more recent work: Strzepek et al. 2019 doi: 10.1073/pnas.1810886116

Line 553: See comment above - I do not think you can solely use Chl-a concentrations to infer reduced light absorption.

Line 559: Your statements above do not agree with this. Here you state clearly mechanisms that help to maintain light absorption - but above you state there is less light being absorbed due to less Chl-a.

Line 564: This statement is false. Ik is the inflection point of light-limiting versus light-saturating. Under Fe-deficiency Ik was higher, indicating that it took more light for the cells to become saturated.

Line 579: I would recommend also calculating the normalised stern-volner NPQ as well and determining whether any differences can be seen here. NPQ stern-volner has been show to have poor correlation with other photophysiological metrics, whereas the normalised stern-volner examines changes in both the dark and light-regulated states. This makes it useful to comparing samples under different Fe conditions. See Schuback et al. 2021 Frontiers for more details.

6. PLOS authors have the option to publish the peer review history of their article (what does this mean?). If published, this will include your full peer review and any attached files.

Reviewer #1: No

Reviewer #2: **Yes: **Thomas Ryan-Keogh

---

## [Editor Report · Decision Letter 1]

15 Nov 2021

The Southern Ocean diatom Pseudo-nitzschia subcurvata flourished better under simulated glacial than interglacial ocean conditions: combined effects of CO2 and iron.

PONE-D-21-25243R1

Dear Dr. Koch,

We’re pleased to inform you that your manuscript has been judged scientifically suitable for publication and will be formally accepted for publication once it meets all outstanding technical requirements.

Kind regards,

Douglas A. Campbell, Ph.D.

Academic Editor

PLOS ONE

Additional Editor Comments (optional):

Nice work!
---

## [Editor Report · Acceptance letter]

17 Nov 2021

PONE-D-21-25243R1 

The Southern Ocean diatom *Pseudo-nitzschia subcurvata* flourished better under simulated glacial than interglacial ocean conditions: combined effects of CO_2_ and iron 

Dear Dr. Koch:

I'm pleased to inform you that your manuscript has been deemed suitable for publication in PLOS ONE. Congratulations! Your manuscript is now with our production department. 

Kind regards, 

on behalf of

Dr. Douglas A. Campbell 

Academic Editor

PLOS ONE